# Implementing a Teacher-Focused Intervention in Physical Education to Increase Pupils' Motivation towards Dance at School

**Diana Amado** [1,*] , **Pablo Molero** [2] , **Fernando Del Villar** [1] , **Miguel Ángel Tapia-Serrano** [2] **and Pedro Antonio Sánchez-Miguel** [2,*]

1   Centre for Sport Studies, Physical Education Area, Rey Juan Carlos University, Alcorcón, 28922 Madrid, Spain; fernando.delvillar@urjc.es
2   Department of Didactic of Musical, Plastic and Corporal Expression, Faculty of Teacher Training, University of Extremadura, 10003 Cáceres, Spain; pmolero@unex.es (P.M.); matapiase@unex.es (M.Á.T.-S.)
*   Correspondence: diana.amado@urjc.es (D.A.); pesanchezm@unex.es (P.A.S.-M.); Tel.: +34-927-257-049 (P.A.S.-M.)

**Abstract:** A teacher-focused intervention that supports the needs for autonomy, competence and relatedness was designed and implemented, to help experienced teachers develop a motivational style during dance teaching sessions at school. Four schools in Mexico, with 12 physical education teachers and 921 pupils, participated in the research. A program was developed at the beginning with the teachers in the experimental group to support the psychological needs of autonomy, competence and relatedness. Both groups were assessed at the beginning and at the end of the program and the results showed that participants from the experimental group had an increase in their perception of autonomy, relatedness and self-determination levels towards dance teaching sessions at school compared with participants from the control group. In conclusion, teachers' training is important to increase pupils' motivation towards dance. Schools should focus on encouraging teachers' "training in motivational strategies to create pupils'" adaptive behaviors.

**Keywords:** teacher training; needs support; motivation; dance; school

---

## 1. Introduction

Dance, in a scholarly context, requires a structure based on scientific principles, and so this investigation puts forward a conceptual framework to articulate methodological practices, from the culture of the movement to expressive activities and practices at all levels. Problems that teachers encountered when they taught this subject were that they were afraid to deal with pupils' negative attitudes and their recognition of the lack of training in this content for teachers [1,2], which has led to proposals that were poorly adapted to the needs of pupil learning and did not improve their motivation [3].

There is currently increasing interest in the study of students' motivation in the educative psychology context [4–6], concerning the usefulness that teachers give to different subjects that are taught in schools [7,8] and, specifically, because of the experiences associated with the different emotions that dancing promotes [9], as one of the more emergent physical activities [10].

When talking about motivational processes within an educative context, the type of motivation that has been related to commitment is the intrinsic motivation [11,12], which is a concept that comes from self-determination theory [13,14]. It is associated with behaviors promoted by the pleasure and satisfaction derived from individual willingness to participate in an activity, which leads to positive consequences such as psychological welfare, interest, enjoyment and intention to persist [14].

Self-determination theory indicates that human behaviors are voluntary or self-determined and reveals that motivation is a multidimensional construct, evolving through a self-determination continuum which includes different types of motivation that have differing degrees of self-determination [13]. These types of motivation are as follows: intrinsic motivation (previously indicated); amotivation, where a person lacks intrinsic and extrinsic motivation towards an activity and does not know exactly why to keep continuing [15]; and extrinsic motivation, which is divided into four types of regulation that vary depending on the degree of self-determination. Thus, from a higher to a lower level of self-determination are the following: integrated regulation (conducts that contribute to defining what one is); identified regulation (behaving in a way that is felt to be personally relevant); introjected regulation (behaving to avoid a sense of guilt or to prove something); and the least self-determined form of regulation is external regulation (behaving to avoid punishment or gain a reward) [13]. Lastly, the lowest level of self-determination is amotivation, in which the subject does not know exactly why they should keep doing an activity.

Furthermore, this perspective indicates that the self-determination level of a person is directly associated by the satisfaction of three basic psychological needs [13]: autonomy, which consists of feelings of volition and the sense that the individual has personal control over his or her own behavior; competence need, characterized by the development of effective interactions and a sense of accomplishment in the context; and relatedness need, explained by the feeling of connection and development of belongingness within the social context in which the activity is developed. Within the context of physical education, it was found that students who considered themselves to be more autonomous, competent and had greater relatedness developed more intrinsic practice reasons [16,17] and positive consequences related to higher effort [18,19], enjoyment and pleasure [20,21], or the increase of physical activity [22,23]. However, several studies have highlighted that the thwarting of basic psychological needs is associated with non-self-determined motivation and maladaptive consequences on health and well-being [24–27].

Therefore, the following question is emphasized: which determinant satisfies individuals' needs for autonomy, competence and relatedness? In accordance with this aspect, the Vallerand's model [28] was developed to better explain the relationship between the variables suggested by self-determination theory. This model gives an explanation about the existence of social factors that play an important role in the satisfaction of these needs [13]. The interpersonal style used by the teacher plays an important role in students' self-determined motivation through the satisfaction of basic psychological needs [29–35].

As was previously indicated, the use of a multidimensional teaching style (based on autonomy, competence and relatedness support), focusing on the model carried out by Skinner and Belmont [36], which was based on including several strategies or skills to support these needs in the teaching style. Specifically, as Tessier et al. [35] highlight, "Autonomy support refers to behaviors by a person in position of authority that show respect, allow freedom of expression and action, and encourage subordinates to attend to, accept, and value their inner states, preferences, and desires" [37]. Teachers can use specific autonomy support strategies, referring to the use of cognitive teaching styles, to give responsibility for selecting tasks that the pupils perform, as well as allowing the volition and the acknowledgment of the pupils' perspectives [38].

Competence support is "when a social context is structured, predictable, contingent, and consistent" ([35], p. 243). The strategies are based to promote students' perceptions of skill, providing positive feedback and recognizing their efforts and progress [36,39].

Lastly, relatedness support "refers to individuals' opportunities to feel related and belonging when they interact within a social environment that offers affection, warmth, care, and nurturance" ([35], p. 243). The strategies are based on creating learning contexts that develop the sense of inclusion, integration, trust and respect among peers [39]. Moreover, the teacher is sympathetic, warm, affectionate and he/she dedicates time, energy and affection to their students [35,40].

Focusing on this idea, the current study has been carried out within the Learning with Dance program, developed by the ConArte (International Consortium of School and Art) organization in "La Nana, Factory of creation and innovation", Cuauhtémoc, Mexico City. This is an initiative inspired by teachers, communicators, artists, cultural promoters and businesspeople, which works through the arts and combines the efforts of different public and private initiatives. It has an international character and looks to contribute to the improvement of education through the development of new capabilities and learning devices. Although it recognizes that all children have artistic skills, the aim of ConArte is not to train artists but to support the development of different types of intelligence that pupils need to manage in an exigent world, therefore contributing to stronger social involvement.

Within this organization, the Learning with Dance program has, since 2006, looked after more than 3000 pupils from public schools in México, integrates dance into the curriculum and involves a large number of agents (pupils, teachers, schools, etc.), which gives it great relevance as one of the few education through art programs that exists in the world, and because of this issue, it has been selected to be the focus of this study. Moreover, the program is currently engaged in a process of revision and methodology systematization, which helps the proposals for collaboration and intervention and promotes further teaching actions.

Therefore, once the study is contextualized, the aim is to analyze the effect produced by a multidimensional intervention program developed by teachers, which incorporates the application of teaching skills directed towards autonomy, competence and relatedness support, on pupils' basic psychological needs satisfaction and self-determined motivation during dance teaching sessions at school. Based on this objective, the hypothesis is that the application of the multidimensional intervention program will produce an increase in the satisfaction of basic psychological needs and self-determined motivation of pupils towards dance teaching sessions at school.

## 2. Materials and Methods

### 2.1. Participants

The sample was formed by 921 pupils, both female ($n = 421$) and male ($n = 500$), ranging in age from 11 to 17 years old ($M = 13.17$ years, $SD = 1.12$), belonging to 4 schools in Mexico City, as well as 12 teachers. Pupils were in the 1st (11–13 years), 2nd (13–15 years) or 3rd (15–17 years) grades. The groups were randomly divided into 24 experimental groups (these were their natural, intact class), with a total of 447 pupils, and 16 control groups (these were their natural, intact class), with a total of 474 pupils.

### 2.2. Instruments

*Perception of basic psychological needs satisfaction*. Satisfaction of students' basic psychological needs satisfaction was measured through a version of the Basic Psychological Needs Measurement Scale [41] translated into Spanish [42]. In the validation of this instrument with a Spanish population, the results of the Confirmatory Factor Analysis (CFA) showed acceptable adjustment values: $\chi^2/gl = 3.29$; CFI = 0.94; IFI = 0.94; TLI = 0.92; RMSEA = 0.07; SRMR = 0.07. Furthermore, the results of the analysis of internal consistency revealed alpha values of 0.81 for autonomy, 0.78 for competence and 0.84 for relatedness. This scale was adapted by changing the wording of the initial sentence, applying it to the context of dance education. Thus, the instrument was preceded by the heading, "In the dance sessions that we have taken part in with our teacher ... ", followed by 12 items (four per factor), which assess the satisfaction of autonomy (e.g., "The way the exercises are carried out correlates perfectly with the way in which I want to do them"), competence (e.g., "I feel that I have progressed greatly with respect to the final objective that I had set out for myself") and relatedness (e.g., "I feel very comfortable when I carry out the exercises with my classmates").

This instrument was tested with a factorial validity analysis with the pre-test scores ($\chi^2/gl = 3.60$; CFI = 0.97; IFI = 0.97; TLI = 0.96; RMSEA = 0.05 and SRMR = 0.03) and post-test scores ($\chi^2/gl = 4.59$;

CFI = 0.96; IFI = 0.96; TLI = 0.95; RMSEA = 0.06 and SRMR = 0.03), showing acceptable adjusted values in both tests, which certified the internal validity of the instrument.

*Motivation*. The Questionnaire on Motivation in Dance and Body Expression [43] was used to assess motivation. This instrument is headed by the statement: "I have participated in the dance sessions that we have taken part in with our teacher . . . ", followed by 20 items grouped into five factors (four per factor) that measure intrinsic motivation (e.g., "Because they are fun"), identified regulation (e.g., "Because I can learn skills that I could use in other areas of my life"), introjected regulation (e.g., "Because it is what I must do to feel good"), external regulation (e.g., "Because it is well looked upon by the teacher and my classmates") and amotivation (e.g., "But I do not understand why we have to have these contents in Physical Education").

This instrument also showed an adequate factorial validity in the pre-test scores ($\chi^2/gl$ = 3.92; CFI = 0.91; IFI = 0.91; TLI = 0.90; RMSEA = 0.06 and SRMR = 0.06.) and post-test scores ($\chi^2/gl$ = 4.59; CFI = 0.96; IFI = 0.96; TLI = 0.95; RMSEA = 0.06 and SRMR = 0.03), showing acceptable adjusted values in both tests.

The answers to both of these instruments were recorded on a 5-point Likert-type scale, where 1 corresponded to "strongly disagree" and 5 corresponded to "strongly agree" with the formulation of the question.

## 2.3. Procedure

*Ethical rules*. The study previously obtained the approval of the Ethics Committee of the University. All participants were treated in agreement with the ethical guidelines of the American Psychological Association. All participants were informed about the aims of the study. Researchers contacted all possible participants. Once permission was obtained from the head teachers of the schools, an informed written consent from the parents was obtained on behalf of the child participants involved in the study.

*Teachers' training program in contents*. Firstly, a training program was developed with teachers (Table 1). The program lasted for 35 h and 12 teachers participated. These teachers belonged to the Learning with Dance program in Mexico. The program included a didactic unit of 12 lessons. Each lesson explained in depth all the contents and applications (see Table 2). Each practical part was developed by one of the 12 teachers. These teachers were responsible for developing a practice class for the other assistants, who had been previously trained and supervised by the principal researcher.

**Table 1.** Contents of the training program for teachers.

| | Contents |
|---|---|
| | Dance and corporal expression at the school |
| | Concept of educational dance |
| | The style technique |
| | The creative technique |
| | Movement factors |
| | Ways of corporal expression |
| | The literacy process of body language |
| **Theme 2** | Body |
| **Theme 3** | Weight |
| **Theme 4** | Contact |
| **Theme 5** | Space |
| **Theme 6** | Time |
| **Theme 7** | Intensity |
| **Theme 8** | Interaction |
| **Theme 9** | Methods for learning to create |
| **Theme 10** | The construction of the observatory's sight |
| | Didactics of the physical-artistic activities |
| | 11.1 Selection of proposals |
| | 11.2 The group dynamic, the constructivist reflection and the evaluation |

**Table 2.** Organization of the programs.

| WEEK 1 | | | | | |
|---|---|---|---|---|---|
| **Hour** | **Monday** | **Tuesday** | **Wednesday** | **Thursday** | **Friday** |
| 09:00–11:00 | Content 1 | Content 3 | Content 5 | Content 7 | Review of Contents |
| 11:00–11:30 | Break | Break | Break | Break | Break |
| 11:30–14:30 | Content 2 | Content 4 | Content 6 | Content 8 | Review of Contents |
| **WEEK 2** | | | | | |
| **Hour** | **Monday** | **Tuesday** | **Wednesday** | **Thursday** | **Friday** |
| 09:00–11:00 | Content 9 | Content 11 | Strategy Training | Strategy Training | Strategy Training |
| 11:00–11:30 | Break | Break | Break | Break | Break |
| 11:30–14:30 | Content 10 | Review of Contents | Strategy Training | Strategy Training | Strategy Training |
| 16:30–20:30 | | | Observer Training | Observer Training | Observer Training |

*Intervention program with teachers to incorporate skills in their teaching*. Once the teachers received the contents, six of twelve teachers were randomly assigned to the experimental groups, developing with them an intervention program based on incorporating skills into their teaching style to increase pupils' motivation. The other 6 teachers were assigned to the control groups and only had to apply the content. The teachers in the experimental group attended a seminar on self-determination theory, which stressed the importance of autonomy, competence and relatedness support. Subsequently, they were given a document in which all of the strategies were explained. Several sessions were used to provide examples of how these strategies may be pedagogically applied (see Table 2). The strategies were taken from Amado [44], following self-determination theory-based recommendations, where each strategy for autonomy, competence and relatedness support appears adapted to the dance context. These strategies are explained below.

Autonomy support: Use the democratic leadership style to foster pupils' active participation, trying not to control or put too much pressure in order to reduce competitiveness; make it possible for them to choose some tasks; and use rewards related to practice and help students to develop assertiveness and efficient communication skills.

Competence support: Adapt activities to students in order to be successful; design activities where success is evaluated through intrapersonal instead of interpersonal indicators; give positive feedback so students feel more secure and more aware of the successes they achieve; and promote aims and feedback regarding the process and not the outcome.

Relatedness support: Include some moments during the lesson to establish personal relationships among students; create mixed-gender groups; foster social skills so that they learn to show empathy towards each other and towards the teacher; and create cooperative exercises to promote the exchange of opinions.

*Observers' training program to supervise the correct application of the teachers intervention with students*. To ensure the adequate implementation of the motivational intervention program, a training program with seven observers was developed (see Table 2). These observers were dance degree students who were interested in the study to have more knowledge in the dance education domain. The fidelity procedure will be explained in detail below.

With the aim that all the strategies were applied, some of them were straight included in the design of activities, for example, adapt exercises to the students' level, design tasks and exercises where success is evaluated through intrapersonal values, give enough time during classes to establish personal relationships and form mixed-gender groups or create cooperative exercises. The rest of the strategies should be presented during the development of all activities of the session, and so, the teacher must focus on them, for example, use democratic leadership, make it possible for students to choose some activities by specifying the norms, use rewards related to practice and place emphasis on pupils' social skills, provide positive feedback and provide objectives and feedback regarding the process and not the outcome. The main researcher met with teachers before giving each session, and they decided the strategies to focus on regarding the moment and the activities in the session, using practical examples to clarify the right form to develop them with students.

*Application of the motivational intervention program with students.* A quasi-experimental design was used, with 40 natural groups already established by the schools, so randomization was not possible. These groups were separated into 16 control groups (*n* = 474) and 24 experimental groups (*n* = 447). Both groups received the same dance teaching program, but in the experimental groups, a motivational intervention program was also carried out. The study was developed throughout the first semester of secondary education (which groups 1st, 2nd and 3rd grade), with twice-weekly 50-min sessions, which took place in the gymnasium of the school, distributed throughout the week (Figure 1).

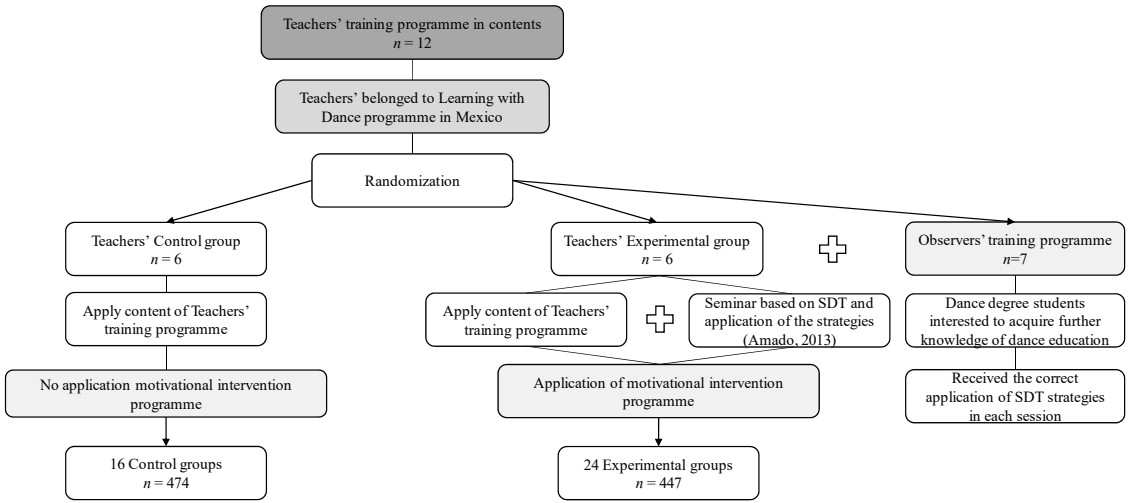

**Figure 1.** Flow diagram of the teachers' training program.

### 2.4. Data Collect

To collect the research data, two measurements were developed. The pre-assessment was carried out during the first class for all groups, and the post-measurement was taken at the end of the process in both groups. Data collection consisted of administering a questionnaire to pupils, and they completed this in the classroom without the presence of the teacher for 20 min approximately. The principal researcher was present at all times to explain any issues and make sure that the process was followed strictly.

### 2.5. Fidelity Procedure

Firstly, the observational sheets were elaborated on the different strategies that must be applied in each session, and they were revised by experts in dance and sports psychology.

Then, a training program with seven observers was developed. It was similar to the program that teachers received in order to incorporate the strategies of autonomy, competence and relatedness support into their teaching style. This procedure lasted five hours, and it was based on a seminar where the concepts of self-determination theory were explained and practice strategies for the application of the different motivational strategies were developed.

Later, the observational sheets for each session were delivered to each observer. In these observational sheets were the session number and contents, and also a list with all the strategies they must apply, in a dichotomous scale. The observers had to write down "yes" or "no", depending on the application or non-application of each strategy by the teacher.

The observational process was performed "live" by observers, who attended each session with the corresponding observational sheet, after a training and contrast process with the members of the research group, based on the observational sheets. For inter-rater reliability, we have used Cohen's kappa coefficient [45]. This coefficient was 0.63, indicating that the agreement between observers is satisfactory according to the Fleiss scale (scores between 0.61 and 0.80 indicate satisfactory agreement; [46]).

This was the form to include a manipulation check to verify that the motivational program was applied correctly. The observers had to rater score the need-supportive motivating style applied by the teachers. When all the scores were "yes", the motivational strategies were applied correctly by teachers and when observers detected a problem in the application of the strategies or the lack of application of any of them, they met at the end of the session with the teacher to ensure a correct implementation of the strategies at the next session. In this way, the observational sheets showed that from the first or second session (until teachers were accustomed to include the motivational strategies in their teaching style), all of them applied the strategies correctly in the sessions.

### 2.6. Data Analysis

The SPSS 18.0 statistical program was used. Firstly, different tests were conducted to determine the nature of the data. The K-S test for independent samples to verify the normality of the groups, the Rachas test for randomness and Levene's test for homoscedasticity or equality between variances were conducted.

Later, a multivariate analysis of variance (MANOVA) was first developed with the pre-test data collected, to examine if there were statistically significant differences in the dependent variables between the two groups (control and experimental) prior to the intervention and, therefore, to check if the two groups were homogenous or heterogeneous. Then, taking into account the differences in the pre-test scores between groups, an ANCOVA (analysis of covariance), with the aim of separating the effect of the intervention program from the differences in the selection of individuals, was developed.

In accordance with the validity of the instruments, an adequate structure for each scale through a factorial analysis was obtained. Moreover, in the reliability analysis, all factors obtained a Cronbach alpha level over 0.70, which is considered adequate.

## 3. Results

### 3.1. Descriptive Statistics and Reliability Analysis

Table 3 shows the descriptive statistics of the variables with the data collected from the pre-test and post-test. The scores obtained in the pre-test and the post-test were similar, with lower values in the post-test in all study variables. The average score was high, ranging between 2.59 and 3.90.

**Table 3.** Descriptive statistics and reliability analysis of the variables included in the research.

| VARIABLES | PRE | | | | | POST | | | | |
|---|---|---|---|---|---|---|---|---|---|---|
| | M | SD | $\alpha$ | Skewness | Kurtosis | M | SD | $\alpha$ | Skewness | Kurtosis |
| **BPN satisfaction** | | | | | | | | | | |
| Competence | 3.79 | 0.84 | 0.71 | −0.69 | 0.26 | 3.50 | 0.91 | 0.74 | −0.48 | −0.09 |
| Autonomy | 3.52 | 0.96 | 0.77 | −0.52 | −0.32 | 3.33 | 0.99 | 0.77 | −0.31 | −0.48 |
| Relatedness | 3.82 | 0.92 | 0.76 | −0.69 | −0.12 | 3.48 | 0.98 | 0.77 | −0.36 | −0.48 |
| **Type of motivation** | | | | | | | | | | |
| Intrinsic | 3.90 | 0.93 | 0.80 | −0.89 | 0.30 | 3.60 | 0.99 | 0.80 | −0.58 | −0.20 |
| Identified | 3.87 | 0.88 | 0.76 | −0.73 | −0.03 | 3.51 | 0.99 | 0.82 | −0.49 | −0.24 |
| Introjected | 3.36 | 0.87 | 0.68 | −0.22 | −0.52 | 3.19 | 0.90 | 0.68 | −0.22 | −0.31 |
| External | 3.62 | 0.87 | 0.70 | −0.42 | −0.34 | 3.33 | 0.92 | 0.69 | −0.24 | −0.38 |
| Amotivation | 2.59 | 0.99 | 0.70 | 0.28 | −0.59 | 2.75 | 0.99 | 0.70 | 0.13 | −0.57 |

### 3.2. Preliminary Analysis of the Groups before Development of the Intervention Program

To begin with, a MANOVA was performed with the pre-test data collected from the dependent variables (perception of basic psychological needs satisfaction and the type of motivation) and included the group as an independent variable (control group and experimental group), to test if the two groups were homogenous before developing the intervention program. The results of this analysis reflected that no statistically significant differences were found at the multivariate level (F (3, 36) = 1.22, p = 0.26; η2 = 0.01).

### 3.3. Comparison of Groups Based on the Effects of the Motivational Intervention Program

An ANCOVA (analysis of covariance) was used with the aim of separating the effects of the intervention program from the differences in the selection of individuals. Thus, with the purpose of examining the intervention program's efficacy, an analysis of variance on the post-tests was conducted, including participants' scores in the pre-test as a covariate (Table 4).

**Table 4.** Analysis of covariance (ANCOVA) of the results of the intervention program (post-test) for the experimental and control groups.

| | Post-Test | | | | | |
| | Experimental Group | | Control Group | | | |
| Measure | M | SD | M | SD | F | $\eta^2$ |
|---|---|---|---|---|---|---|
| **BPN satisfaction** | | | | | | |
| Competence | 3.54 | 0.87 | 3.46 | 0.96 | 2.21 | 0.00 |
| Autonomy | 3.40 | 0.99 | 3.25 | 0.99 | 6.95 ** | 0.01 |
| Relatedness | 3.53 | 0.98 | 3.42 | 0.98 | 6.51 ** | 0.01 |
| **Type of motivation** | | | | | | |
| Intrinsic motivation | 3.65 | 0.95 | 3.54 | 1.04 | 3.60 * | 0.00 |
| Identified regulation | 3.56 | 0.4 | 3.45 | 0.5 | 3.44 * | 0.00 |
| Introjected regulation | 3.22 | 0.88 | 3.17 | 0.92 | 0.73 | 0.00 |
| External regulation | 3.36 | 0.88 | 3.30 | 0.97 | 1.66 | 0.00 |
| Amotivation | 2.74 | 0.96 | 2.74 | 1.03 | 0.18 | 0.00 |

Note. $p \leq 0.01$ **; $p \leq 0.05$ *.

Firstly, the results found for pupils' perceptions of basic psychological needs satisfaction, after controlling the "autonomy satisfaction pre-test" and "relatedness satisfaction pre-test" (covariate) levels, a significant effect of the intervention on the "autonomy satisfaction post-test" ($F_{(1, 38)} = 6.95$, $p = 0.01$, $\eta2 = 0.01$) and "relatedness satisfaction post-test" ($F_{(1, 38)} = 6.51$, $p = 0.01$, $\eta2 = 0.01$). Participants who received the intervention (experimental group) significantly increased their perception of autonomy and relatedness satisfaction at the end of the program, compared with the control group which did not receive the intervention. However, no significant effects of the intervention were obtained for the "competence satisfaction post-test" ($F_{(1, 38)} = 2.21$, $p = 0.14$, $\eta2 = 0.00$) after controlling the "competence satisfaction pre-test", so participants in the experimental group perceived no changes to their perception of competence satisfaction at the end of the intervention program, compared with the control group which did not receive the intervention.

With regards to the type of motivation, significant differences were found in intrinsic motivation ($F_{(1, 38)} = 3.60$, $p = 0.05$, $\eta2 = 0.00$) and identified regulation ($F_{(1, 38)} = 3.44$, $p = 0.05$, $\eta2 = 0.00$), so participants who received the intervention (experimental group), significantly increased their intrinsic motivation and their identified regulation at the end of the program, compared with the control group which did not received the intervention. On the other hand, no significant differences were found in introjected regulation ($F_{(1, 38)} = 0.73$, $p = 0.39$, $\eta2 = 0.00$), external regulation ($F_{(1, 38)} = 1.66$, $p = 0.20$, $\eta2 = 0.00$) and amotivation ($F_{(1, 38)} = 0.18$, $p = 0.67$, $\eta2 = 0.00$). Thus, participants in the experimental group perceived no changes to their introjected regulation, external regulation and amotivation at the end of the intervention program, compared with the control group which did not receive the intervention.

## 4. Discussion

The aim of this research was to test the effect of a multidimensional intervention program, based on the application of teaching skills directed towards autonomy, competence and relatedness support, on the satisfaction of pupils' basic psychological needs and their self-determined motivation during dance teaching classes.

Firstly, regarding the satisfaction of basic psychological needs, the results showed that pupils who belonged to the experimental group had significantly greater perceptions of autonomy and relatedness after the application of the program with respect to the control group that did not receive any program. Previous studies have also found that pupils who perceive support for basic psychological needs from the teacher during physical education sessions show greater needs satisfaction [19,26,27,47].

These results emphasize the importance of the teacher's role in the teaching of dance at school, and therefore they should include in their teaching style strategies to support pupils' autonomy, competence and relatedness. Strategies to support autonomy show the use of a democratic leadership style, encourage pupils' active participation, make them feel important within the group, provide choices and options, provide some freedom and responsibility when it comes to performing tasks, try not to control or put too much pressure on pupils, explain the objective of the activity, use rewards related to practice and place emphasis on pupils' social skills [38,44].

For competence support, some examples of strategies are as follows: adapting exercises to the pupils' level by balancing the difficulty and capacity; give positive feedback; provide objectives and feedback regarding the process, and not the result, by acknowledging pupils' efforts and/or improvements; and promote aims at the short-, medium- and long-term, with the aim that they clearly understand their purposes and progress towards them, with sufficient time provided for all pupils to achieve the set objectives [36,39,44]

Finally, with respect to interpersonal involvement, some of the strategies are as follows: form mixed-gender groups to boost the integration of all students by changing the criteria for the creation of groups, group dynamics, role plays or trust activities, in order to improve all students' feelings of belonging within the group; and educate students in social skills so that they learn to show empathy towards each other and towards the teacher [35,39,40,44].

It is important to note that, although teaching support for basic psychological needs has produced a slight increase in the perceptions of autonomy and relatedness in the experimental group, it did not promote significant changes in the perceptions of pupils' competence. This suggests that pupils who received the motivational intervention program did not increase their perception of competence, when compared with pupils from the control group that did not receive that program. Similar results were found by Amado et al. [48] in the context of dance in physical education, and Tessier et al. [35] using different sports in physical education, who at the end of the intervention program pointed out the lack of significant changes in the perceptions of competence, emphasizing that time is needed when a significant change to the beginning of new tasks is required, and the length of the program may not have been enough to improve pupil competence because this is the need that requires the most time to be satisfied, as it requires changes in the individual perceptions of personal skills. Hence, it is probably the case that in the current study, the length of the program was also not enough because when greater autonomy is given to pupils (as happened in the experimental group) in the learning of new topics such as dance, perceptions of skill are low, and it might be necessary to have more time to practice the specific skill before seeing improvements in pupils' perceptions of competence [49–51].

Secondly, focusing on the self-determination level, the results highlighted an increase in intrinsic motivation and identified regulation in the experimental group, when finishing the program, compared with the control group, which suggests that the motivational intervention program increased pupils' self-determined motivation, so they felt more satisfaction and pleasure during the practice of dance in the school and were more likely to identify it as something important to them [13].

In accordance with these results, several researchers have shown that the use of a teaching style, focused on the satisfaction of basic psychological needs by teachers, has important relevance for the students' self-determined motivation [27,30,31,34]. On the contrary, other authors such as Amado et al. [48], Ntoumanis et al. [52] and Tessier et al. [35] have indicated that physical education teacher support for basic psychological needs did not cause a significant increase in self-determined motivation.

Previous studies have found a relationship between teaching style and pupils' self-determined motivation during physical education classes [27,29,30,53–57], but it is important to note that these works only measured the teacher support style regarding autonomy in a one-dimensional way, instead of assessing the impact of a three-dimensional style: autonomy, competence and relatedness support, as has been performed in the present research.

Some limitations arise from the study. Firstly, it is important to note that in the Learning with Dance program, classes were simultaneously taught by three teachers, two specialists in dance and one music teacher, which made it much more difficult to implement teacher training and coordination between them, in order to apply the motivational strategies. Nevertheless, observers were responsible for the control of the application of the program, and previously trained for it. Here, the second limitation emerged, which was that some of the observers that assisted in the initial training program could not assist in the classes due to personal reasons, and so we had to delete the data for the groups that these observers belonged to. Finally, it is important to note that the teachers involved in the application of the program (experimental group) had such a large amount of responsibility during the process that they were often ready to drop out of the investigation and go back to teaching their normal classes. In the end, this did not happen because we retained the unconditional support of ConArte, which helped us to encourage and motivate the teachers to continue with the commitment they had given to us at the beginning.

In accordance with the limitations found, in the future, we propose to expand teachers' training and spread it out over time, in the previous months to the application of a program of these characteristics. This could improve the understanding of the concepts to conduct an adequate application of the motivational strategies, avoiding problems of unsafety in teachers as a result of the program demands. Another possibility for further studies to fulfill the aims indicated at certain times during the teaching performance would be the necessity of methodological training according to their reality, creating measurement instruments for this purpose. Moreover, the most appropriate issue is that responsibility for the application of strategies rests with a single teacher, although the other teachers have knowledge about it, which helped with the training and later coordination between the teachers. Lastly, regarding observers, at the beginning of the study, a questionnaire should have been conducted to find out their personal details, such as if they have another job or if they live in the same city, among others, accompanying the signed commitment to participate in the study.

## 5. Conclusions

After testing the results, which partially confirm the hypothesis of the study, it has been shown that the application of a motivational intervention program with physical education teachers produces an increase in the basic psychological needs satisfaction and pupils' self-determined motivation towards teaching dance at school. The findings showed the utility of an intervention program of 12 teaching classes of dance in school, based on the inclusion of teaching skills to support pupils' autonomy, competence and relatedness, for increasing perceptions of autonomy, relatedness and self-determined motivation among pupils. However, this program was not enough to increase pupils' perceptions of their own competence during dance classes, which makes it necessary to continue research into teaching procedures that increase perceptions of competence in pupils.

## 6. Contribution to the Scientific Community

Using the quasi-experimental research design, the present study replicates and expands upon previous self-determination-based intervention studies in school settings. Our findings reinforce the usefulness of the three-dimensional approach in the field of educational dance to understanding teachers' interpersonal styles and its effects on student motivation. The results emphasize the importance of the teacher's role in the teaching of dance at school, and therefore they should include in their teaching style strategies to support pupils' autonomy, competence and relatedness. In this study, each of the strategies applied to dance teaching is presented in detail

and it is recommended how they should be carried out, either in the design of the activities or during the implementation of the session. Moreover, the whole design is shown, with the aim to give further information to researches who aim to replicate motivational interventions programs in educational contexts.

**Author Contributions:** Conceptualization, D.A.; methodology, D.A.; software, D.A. and P.A.S.-M.; validation, D.A., P.A.S.-M. and F.D.V.; formal analysis, D.A. and P.M.; investigation, D.A., P.A.S.-M., P.M. and F.D.V.; resources, D.A., P.M. and M.Á.T.-S.; data curation, D.A.; P.A.S.-M. and M.Á.T.-S.; writing—original draft preparation, D.A.; writing—review and editing, D.A. and F.D.V.; visualization, M.Á.T.-S.; supervision, F.D.V.; project administration, P.A.S.-M. All authors have read and agreed to the published version of the manuscript.

**Funding:** This study did not receive funding.

**Acknowledgments:** The authors would like to thanks ConArte for trusting our work and inviting us to develop the training and intervention program and for all the aids provided during the process. The work has been done as part of a multidisciplinary training program for the promotion of dance at school. The authors wish to thank the schools, teachers and students who participated in the research.

**Conflicts of Interest:** The authors declare no conflict of interest.

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
