# Peer review of "Implementing a Teacher-Focused Intervention in Physical Education to Increase Pupils’ Motivation towards Dance at School"

_sustainability, doi:10.3390/su12114550_

Round 1

Reviewer 1 Report

The study presents an interesting proposal in the field of motivation and physical education. Despite this, a series of recommendations are made below to improve the manuscript:

  • References must be updated. Only 4 references are from the last 5 years.
  • It is indicated that the Basic Psychological Needs Measurement Scale has been used and translated into Spanish, but it does not indicate how the translation process was carried out or whether the scale was previously validated in the Spanish population. Alpha values are also required in questionnaires
  • Check line 163
  • The time frame for implementation of activities by groups and the motivational programme is not clear. I recommend a figure or table in which the times and the different phases (pretest-postest) are visually collected by groups.

Author Response

Dear Millie Huo,

We would like to thank reviewers for their comments and suggestions on the manuscript which have improved the quality of the article.

We hope to have responded correctly all the reviewers` comments.

Dr. Diana Amado Alonso

Author: We appreciate the reviewers` confidence in our research, as well as comments and suggestions indicated about it, because we consider that they have helped to improve the quality of the manuscript. Following, we will explain the changes included in the text according to the indications.

  1. Reviewer 1: The study presents an interesting proposal in the field of motivation and physical education. Despite this, a series of recommendations are made below to improve the manuscript:

References must be updated. Only 4 references are from the last 5 years.
It is indicated that the Basic Psychological Needs Measurement Scale has been used and translated into Spanish, but it does not indicate how the translation process was carried out or whether the scale was previously validated in the Spanish population. Alpha values are also required in questionnaires.

Author: Changes suggested by reviewer has been conducted, as can be seen in the document. Firstly, we have reviewed and updated the references, except those that belong to relevant studies published in prestigious journals.

Regarding the validation of the aforementioned instrument, we have included the CFA data and internal consistency analysis with Spanish population. Please, if you need some more information indicate us.

The time frame for implementation of activities by groups and the motivational programme is not clear. I recommend a figure or table in which the times and the different phases (pretest-postest) are visually collected by groups.

Author: Following reviewers´ suggestion, we have included a figure explaining in detail the intervention program (Please see Figure 1).

Reviewer 2 Report

This document provides a comprehensive review of most recent studies on this particular topic/question. The topic of this paper is relevant, timely, and of interest to the audience of this journal. It is based on rigorous academic standards and its content is technically accurate and sound. The theoretical framework is well established, and a recent literature review is presented. The authors provide extensive and detailed literature review, and make a strong argument for their contribution to basic knowledge in this area.

The research methodology for the study is appropriate and applied properly. The results of analysis are correctly interpreted and conclusions are complete, even if not generalized to other contexts and populations. Overall, I found this study to be original, well grounded in the literature, and well executed. I recommend that it be published with minor editing revisions.

Author Response

Dear Millie Huo,

We would like to thank reviewers for their comments and suggestions on the manuscript which have improved the quality of the article.

We hope to have responded correctly all the reviewers` comments.

Dr. Diana Amado Alonso

Author: We appreciate the reviewers` confidence in our research, as well as comments and suggestions indicated about it, because we consider that they have helped to improve the quality of the manuscript. Following, we will explain the changes included in the text according to the indications.

  1. Reviewer 2: This document provides a comprehensive review of most recent studies on this particular topic/question. The topic of this paper is relevant, timely, and of interest to the audience of this journal. It is based on rigorous academic standards and its content is technically accurate and sound. The theoretical framework is well established, and a recent literature review is presented. The authors provide extensive and detailed literature review, and make a strong argument for their contribution to basic knowledge in this area.

The research methodology for the study is appropriate and applied properly. The results of analysis are correctly interpreted and conclusions are complete, even if not generalized to other contexts and populations. Overall, I found this study to be original, well grounded in the literature, and well executed. I recommend that it be published with minor editing revisions.

Author: Authors would like to thank your words towards our research, as well as the value encouraged towards the document.

Round 2

Reviewer 1 Report

All suggestions have been made correctly